# Monocarboxylate Transporter 1 (MCT1) in Liver Pathology

**DOI:** 10.3390/ijms21051606

**Published:** 2020-02-26

**Authors:** Marek Droździk, Sylwia Szeląg-Pieniek, Justyna Grzegółkowska, Joanna Łapczuk-Romańska, Mariola Post, Pawel Domagała, Janusz Miętkiewski, Stefan Oswald, Mateusz Kurzawski

**Affiliations:** 1Department of Experimental and Clinical Pharmacology, Pomeranian Medical University, 70-111 Szczecin, Poland; sylwia.szelag@pum.edu.pl (S.S.-P.); justyna.bukato@hotmail.com (J.G.); joanna.lapczuk@pum.edu.pl (J.Ł.-R.); mkurz@pum.edu.pl (M.K.); 2Department of General and Transplantation Surgery, County Hospital, 71-455 Szczecin, Poland; mariolapost@wp.pl; 3Department of Pathology, Pomeranian Medical University, 71-242 Szczecin, Poland; paweldom@pum.edu.pl; 4Department of Pathology, Marie-Curie County Hospital, 71-455 Szczecin, Poland; 5Department of Clinical Pharmacology, University Medicine of Greifswald, 17489 Greifswald, Germany; stefan.oswald@uni-greifswald.de; 6Institute of Pharmacology and Toxicology, Rostock University Medical Center, 18051 Rostock, Germany

**Keywords:** monocarboxylate transporter 1, *SLC16A1*, liver pathology

## Abstract

Membrane monocarboxylate transporter 1 (*SLC16A1*/MCT1) plays an important role in hepatocyte homeostasis, as well as drug handling. However, there is no available information about the impact of liver pathology on the transporter levels and function. The study was aimed to quantify *SLC16A1* mRNA (qRT-PCR) and MCT1 protein abundance (liquid chromatography–tandem mass spectrometry (LC¬¬–MS/MS)) in the livers of patients diagnosed, according to the standard clinical criteria, with hepatitis C, primary biliary cirrhosis, primary sclerosing hepatitis, alcoholic liver disease (ALD), and autoimmune hepatitis. The stage of liver dysfunction was classified according to Child–Pugh score. Downregulation of *SLC16A1*/MCT1 levels was observed in all liver pathology states, significantly for ALD. The progression of liver dysfunction, from Child–Pugh class A to C, involved the gradual decline in *SLC16A1* mRNA and MCT1 protein abundance, reaching a clinically significant decrease in class C livers. Reduced levels of MCT1 were associated with significant intracellular lactate accumulation. The MCT1 transcript and protein did not demonstrate significant correlations regardless of the liver pathology analyzed, as well as the disease stage, suggesting posttranscriptional regulation, and several microRNAs were found as potential regulators of MCT1 abundance. MCT1 membrane immunolocalization without cytoplasmic retention was observed in all studied liver pathologies. Overall, the study demonstrates that *SLC16A1*/MCT1 is involved in liver pathology, especially in ALD.

## 1. Introduction

The monocarboxylate transporter 1 (MCT1), encoded by the *SLC16A1* gene, belongs to the monocarboxylate transporter family, being a member of the solute carrier (SLC) transporter superfamily. Based on sequence homology, the MCT family includes 14 members. MCT1 is responsible for the transmembrane H^+^ coupled transport of short chain monocarboxylates, primarily L-lactate, pyruvate, and ketone bodies. MCT1-mediated transport supports the maintenance of an energy balance and pH homeostasis, as well as enabling metabolic cooperation between different tissues with distinct energetic profiles. The transporter facilitates either the influx or efflux of the substrates, depending on substrate and H^+^-concentration gradients [1]. *SLC16A1* is expressed in tissues that rely on the efflux or uptake of energy metabolites, based on the energetic profile (e.g., brain, heart, kidney, lungs, liver, muscle, placenta, and erythrocytes), but also in the gastrointestinal tract to support the absorption of short chain fatty acids, such as acetate, propionate, or butyrate [2,3,4].

There is some information about the role of MCT1 in human healthy tissues and pathological states. The MCT1 role in cancer development and progression is hypothesized. In cancer, a metabolic cooperation between hypoxic glycolytic and oxygenated oxidative neoplastic cells, as well as between oxidative tumor and glycolytic stromal cells, has been documented [5]. The MCT1-mediated efflux of lactates from highly glycolytic tumor/stromal cells ensures the progression of glycolysis and prevents intracellular acidification (extracellular lactates provide not only a respiratory fuel for oxidative cells, but also contribute to an acidic tumor microenvironment promoting migration, angiogenesis, and immunosuppression). A similar metabolic phenomenon is observed in skeletal muscles or in the brain, where MCT1 and other MCTs cooperate in lactate shuttling between various cell populations, where MCT1 provides influx activity in oxidative, slow-twitch red muscle fibers [2], or efflux transport from glycolytic astrocytes [6].

There is little information about MCT1’s presence and its role in liver functions. *SLC16A1* expression in the human liver measured at the mRNA level has been evidenced by Williams et al. [7]. Our studies demonstrated a presence of not only mRNA, but also the MCT1 protein in human livers, in both organ donors as well as metastatic liver disease hepatic tissues [4,8]. Metastatic livers were characterized by a significantly elevated content of *SLC16A1* mRNA transcripts in comparison to donor samples, but comparable abundance of MCT1 protein. In liver parenchymal cells, MCT1 may be used to transport L-lactate into hepatocytes for gluconeogenesis, for which it is a major substrate, especially after exercise [9]. However, there is no available information about *SLC16A1* expression in human liver pathological states. Therefore, we aimed to evaluate the transporter mRNA and protein levels in liver pathologies of different etiologies—i.e., viral (hepatitis C), toxic (alcoholic liver disease), cholestatic (primary biliary cirrhosis, primary sclerosing cholangitis), and inflammatory (autoimmune hepatitis).

## 2. Results

### 2.1. mRNA Quantification

The relative *SLC16A1* gene expression measured at the mRNA level in the studied pathological livers was lower than in the control samples (mean relative mRNA quantity in pooled liver pathological samples was approximately 71% of that in control samples; *p* < 0.01). However, only alcoholic liver disease (ALD) and primary biliary cholangitis (PBC) patients were characterized by significant downregulation of gene expression (*p* < 0.05) when different liver pathologies were analyzed separately (Figure 1, Appendix A). Expression of *SLC16A1* gradually declined with the progression of liver function deterioration, being significantly lower in patients classified as Child–Pugh class C compared to controls (Figure 1, Appendix A).

### 2.2. Protein Abundance

Quantitative changes in *SLC16A1* mRNA transcript were paralleled by quantitative downregulation of MCT1 protein content. The mean content of MCT1 proteins was 115.7 fmol/mg tissue in controls, compared with 58.4 fmol/mg in pooled pathological samples (*p* < 0.01). When analyzing different pathologies separately, the protein abundance of MCT1 was lower in all the studied groups (as compared with controls), but the difference reached statistical significance only in case of ALD samples (Figure 1, Appendix A). Similar to relative mRNA quantity, MCT1 protein levels were lower in all stages of liver dysfunction, but that difference was significant only when Child–Pugh C-stage livers were compared to controls (Figure 1, Appendix A).

### 2.3. mRNA/Protein Correlation

The *SLC16A1* mRNA transcript and MCT1 protein quantities did not demonstrate significant correlation, regardless of the liver pathology or the disease stage analyzed (classified according to the Child–Pugh score) (Appendix A).

### 2.4. Correlation between miRNAs and MCT1

A total number of 35 miRNAs potentially involved in *SLC16A1* regulation (based on database search) were included in miRNA microarray analysis. Among those, 16 distinct miRNAs were present in the analyzed liver samples at the quantitative level, allowing miRNA/protein correlation analysis. Significant negative correlations were documented between miR-27a-3p, miR-145-5p, and MCT1 protein abundance in the control group. In ALD livers, a significant correlation was noted for miR-605-5p (Table 1). As for other liver pathologies, significant negative correlations were detected for miR-27a-3p in primary sclerosing cholangitis (PSC), for miR-29c-3p and miR-342-3p in primary biliary cholangitis (PBC), and for miR-29c-3p, miR-374b-5p, and miR-495-3p in autoimmune hepatitis (AIH). However, none of the observations remained significant after correction for multiple testing (Bonferroni correction). Two miRNAs, miR-27a-3p and miR-342-3p, were significantly inversely correlated with MCT1 in pooled liver pathological samples (*n* = 77). When all the liver samples (*n* = 97) were jointly analyzed for a negative correlation between miRNA and the MCT1 protein, only weak significant correlations (−0.3 < *r* < −0.2) were observed. After application of the Bonferroni correction, the relative quantity of three distinct miRNAs—mir-27a-3p, miR-342-3p, and miR-374a-5p—was significantly inversely correlated with MCT1 protein abundance (Table 1).

### 2.5. Lactate Concentration, pH, and Their Correlation with MCT1 in the Liver

Mean lactate concentration in alcoholic liver disease was significantly higher than in the controls (*p* < 0.01)—i.e., 1.65 ± 0.50 nmol/mg tissue vs. 0.63 ± 0.41 nmol/mg tissue (Figure 2). The pH values in the studied liver tissues (mean ± SD) were 6.71 ± 0.21 for ALD and 6.65 ± 0.17 for control samples, and did not differ significantly. There was a significant positive correlation between tissue lactate concentration and MCT1 abundance in controls, but not in ALD liver samples. On the other hand, a strong correlation was observed between lactate concentration and pH in ALD samples, which was not observed in control livers. The results of correlation analysis between the parameters are presented in Figure 3.

### 2.6. Immunohistochemistry

Positive membranous expression of MCT1 was found in all studied liver pathologies, as well as in the control hepatic samples (Figure 4).

## 3. Discussion

*SLC16A1* expression was confirmed in healthy livers, where the transporter is presumed to provide L-lactate transport into hepatocytes used in turn for gluconeogenesis [10]. However, its role in liver pathology was not defined. Here we report, for the first time, information about *SLC16A1* expression at both the mRNA and protein levels in a wide range of liver pathologies, i.e., of viral (hepatitis C), toxic (alcoholic liver disease), cholestatic (primary biliary cirrhosis, primary sclerosing cholangitis), and inflammatory (autoimmune hepatitis) origin. Decreased levels of *SLC16A1* mRNA and MCT1 proteins were observed in all the studied pathologies, regardless of the etiology, and it was especially pronounced in alcoholic liver disease. Moreover, progression of liver dysfunction, from Child–Pugh class A to C, was associated with a gradual decline in *SLC16A1* expression, reaching a significant decrease in the class C livers.

Based on the postulated role of MCT1 in hepatocytes, it seems that liver pathology entails deficient intracellular transport of lactates, thus reducing the capacity for gluconeogenesis and creating energetic deficiency. The above function was best defined in cancer cells, where MCT1 is engaged in the influx shuttling of lactates into neoplastic cells, providing an energy source for further tumor development [5]. Otherwise, as MCT1 also provides efflux activity in glycolytic and hypoxic cells, its deficient status may contribute to the development of intracellular acidosis [2]. Therefore, in the case of hepatocytes the role of MCT1 can be dual, and related to the cell functional state as well as the underlying pathology. Thus, based on the present study results, decreased MCT1 abundance seems to be directly related to liver pathology development by either insufficient intracellular lactate shuttling (producing energy deficits) or deficient lactate efflux (producing intracellular acidification and functional deterioration of the hepatocyte). Our study revealed significantly higher lactate levels in livers obtained from patients diagnosed with alcoholic liver disease compared to healthy control livers. Those results are in keeping with the report on higher lactate levels in livers in the course of nonalcoholic steatohepatitis (NASH) and alcohol-related liver damage published by Schofield et al. [11].

Our findings suggest that reduced content of MCT1 in ALD may be related to impaired lactate efflux. However, increased lactate efflux did not produce pH changes in ALD compared to the controls, most probably due to intracellular buffer function, e.g., provided by the activity of lactate dehydrogenase [12,13]. The present study may also suggest that intracellular lactate regulates MCT1 protein abundance, as a positive correlation between lactate concentration and MCT1 protein content was observed in healthy livers, but this significant relationship was lost in pathological livers. Thus, deterioration of liver function may involve dysregulation of *SLC16A1* expression, which further leads to intracellular lactate accumulation. However, the regulatory mechanisms of *SLC16A1* expression in hepatocytes are not defined. This study shows that MCT1 levels are not correlated with tissue pH (in either controls or alcoholic disease livers). An effect of pH on MCT1 function in red blood cells was investigated by Morse et al. [14]. The authors revealed that γ-hydroxybutyrate shuttling via MCT1 decreased in acidic environment (from 7.4 to 6.5). If these findings could be applied to the present study, and the pH results from the control and ALD livers, it could be stated that pH did not affect MCT1 activity, and as discussed above, MCT1 protein abundance.

The *SLC16A1* mRNA and MCT1 protein quantities did not demonstrate significant correlation regardless of the liver pathology analyzed or the disease stage (classified according to the Child–Pugh score), suggesting posttranscriptional regulation. In the present study, a potential involvement of miRNA in regulation posttranscription of *SLC16A1* expression was analyzed. From a pool of miRNAs potentially regulating *SLC16A1* expression, 16 distinct transcripts were detected at the quantitative level in the investigated liver samples. Several negative correlations between miRNA and MCT1 protein concentration were observed in the current study.

There is currently no direct information about the involvement of miRNA in MCT1 in human hepatocytes. However, accumulated data from other cells/tissues and experimental models suggests the involvement of this posttranscriptional mechanism in the regulation of MCT1 activity. It was determined that miR-124a-3p and miR-342-3p could be implicated in *SLC16A1*/MCT1 regulation in cancer cells, while miR-29a-3p and miR-29b-3p may be responsible for transporter downregulation in pancreatic β-cells [13,15,16]. In the current study, miR-124-3p was not expressed at a detectable level in the analyzed liver samples, so it is unlikely to play a role in the regulation of *SLC16A1* expression in hepatocytes. In contrast, miR-342-3p was present in liver tissue and was significantly negatively correlated with MCT1 concentration in liver pathologies, as well as in all the study samples jointly analyzed (control and liver pathology). Even though the correlation was not very strong, that observation supports the involvement of miR-342-3p in posttranscriptional silencing of *SLC16A1*. As for miR-29a-3p and miR-29b-3p, those transcripts showed no significant correlation with MCT1 quantity. However, for closely related miR-29c-3p, a significant inverse correlation was noticed in PBC and AIH livers, so the role of miR-29 miRNAs in hepatic regulation of MCT1 needs to be validated in further studies.

Among miRNAs have not been previously investigated in relation to MCT1 expression, only miR-27a-3p showed significant inverse correlation with protein quantity, both in control samples and liver pathologies, as well as in all jointly analyzed tissue samples. This miRNA has been extensively studied before, and was dysregulated in many types of cancer, including hepatocellular carcinoma [17]. Additionally, miR-27a-3p affects drug sensitivity and the resistance of cancer cells, as its upregulation promotes the expression level of *ABCB1*/*MDR1* gene, but it has not been studied in relation to SLC transporters up to date [18]. Another candidate for MCT1 regulation was miR-145-5p, which was significantly correlated with MCT1 content in control samples. That particular miRNA is expressed in hepatocytes, probably acting as a tumor suppressor, and its downregulation may contribute to the pathogenesis of hepatitis B virus-associated hepatocellular carcinoma [19]. Finally, miR-374a-5p, previously investigated mainly as a biomarker in several types of cancer, was significantly inversely correlated with MCT1 concentration in pooled liver samples. However, all the obtained miRNA data must be treated as preliminary, as no functional validation of miRNA involvement in the regulation of *SLC16A1* expression was performed in the present study, and further evidence must be obtained from additional in vitro experiments.

It is known that in some liver pathologies, dysregulation of the membrane transporters trafficking occurs, such as MRP2 intracellular retention due to dysfunctional transporter interaction with its auxiliary protein, i.e., ezrin, in the cholestatic state [20]. In the case of MCT1, CD147 (basigin) is its ancillary protein, determining translocation to plasma membrane [21]. The present study demonstrated MCT1 membrane localization without cytoplasmic retention in all studied liver pathologies regardless of their origin, i.e., toxic, inflammatory, or cholestatic. This finding implies that quantitative measurements of MCT1 in liver dysfunction may instead be directly related to its function.

In conclusion, this study provides information on MCT1 status in a wide range of liver pathologies. The results suggest downregulation of MCT1 with consequential intracellular accumulation of lactates. Induction of MCT1 level/function could be a potential target in liver insufficiency, as in cancer therapy when opposite (downregulation) intervention has been postulated and experimentally tested [22,23]. However, the observations from the present study should be verified in larger-scale observations, as well as by functional validation.

## 4. Materials and Methods

### 4.1. Liver Samples

The control samples from Caucasian subjects were harvested from metastatic colon cancer livers, at a site at least 5 cm distant from the tumor. The patients were aged 63 ± 10 years, with 11 males and 9 females, and were free from chronic diseases (except for one patient with hypertension and prostate hypertrophy medicated with bisoprolol, furosemide, and tamsulosin; one patient with hypertension medicated with bepridil; and one patient with hypertension, medicated with amlodipine). The collected tissues did not show any pathological signs, as confirmed by histological examination. 

The liver pathology samples were dissected from liver parenchymal tissue of patients diagnosed (according to the standard clinical criteria) with hepatitis C (HCV), primary biliary cholangitis (PBC), primary sclerosing cholangitis (PSC), alcoholic liver disease (ALD), and autoimmune hepatitis (AIH) during elective liver transplantation. All patients met clinical criteria for liver transplantation. The patients’ characteristics are presented in Table 2.

Tissue biopsies were dissected from livers (both the control and pathological) under standard general anesthesia (propofol, sevoflurane, rocuronium, fentanyl, dipyrone) not later than 15 min after blood flow arrest. The liver samples were immediately snap-frozen in liquid nitrogen for protein analysis or immersed in RNAlater (Applied Biosystems, Darmstadt, Germany) for RNA analysis, and then stored at −80 °C. The study protocol was approved by The Bioethics Committee of Pomeranian Medical University from 27 February 2007 (approval code BN-001/11/07).

### 4.2. mRNA Quantification

Total RNA was isolated from 40–50 mg of the liver tissue, and quantified via real-time qRT-PCR, as recently described [4]. The following pre-validated TaqMan gene expression assays (Applied Biosystems, Foster City, CA, United States) were used: *SLC16A1* (Hs01560299_m1) and the reference genes *PPIA* (Hs04194521_s1), *RPLP0* (Hs99999902_m1), *RPS9* (Hs02339424_g1), *ACTB* (Hs99999903_m1), and *HMBS* (Hs01560299). Relative *SLC16A1* mRNA was calculated with the ΔΔC_T_ method: normalized to mean expression of the housekeeping genes and the mean value for the control group.

### 4.3. Protein Quantification by LC−MS/MS

The abundance of hepatic transporter protein was quantified by mass spectrometry-based targeted proteomics, as recently described [4]. About 40 mg of pulverized tissue was added to 1 mL lysis buffer (0.2% SDS, 5 mM EDTA) containing 5 µL/mL Protease Inhibitor Cocktail (ProteoExtract-Native Membrane Extraction Kit; Merck, Darmstadt, Germany), and manually homogenized using a Dounce homogenizer (10 strokes) before incubation for 30 min at 4 °C. After determination of the protein concentration (Pierce BCA Protein Assay Kit; Thermo Fisher Scientific, Hennigsdorf, Germany), a volume corresponding to 100 µg protein was subjected to the established method of filter-aided sample preparation (FASP), which generates tryptic digests of whole tissue lysates and avoids potential disadvantages of other sample preparation methods, such as sample loss or the enrichment of certain cell fractions [4,24]. The resulting MCT1 protein data were normalized to the respective mass of tissue lysate used in the tryptic digest. During the analytical period, the accuracy of the method was within ± 25% (relative error), as determined by analyzing quality control samples containing low (0.25 nmol/L), middle (2.5 nmol/L), and high (25 nmol/L) peptide concentrations, which were measured before, within, and after the tissue samples [25].

### 4.4. miRNA Quantification and Analysis

Quantitation of 754 human miRNAs was performed with qRT-PCR microfluidic cards (TaqMan Array Human MicroRNA A+B Cards Set v3.0, Thermo Fisher Scientific, United States) in the same samples of total RNA (500 ng) that were used for mRNA analysis. Reverse transcription was performed using TaqMan MicroRNA Reverse Transcription Kit in two separate reactions, each containing different pool of Megaplex RT Primers (Human Pools A and B). The ViiA 7 Real-Time PCR System and TaqMan Universal Master Mix II (Thermo Fisher Scientific, United States) were used for running real-time PCR reactions in microfluidic cards. A cut-off of 32 for C_T_ values was used, as recommended by protocol from the assay provider. The relative quantity of each miRNA was determined after normalization to the mean expression of three endogenous control RNAs (stably expressed, small, non-coding RNAs: U6 snRNA, RNU44, and RNU48; ΔC_T_ method). A total number of 457 unique miRNAs was detected in liver samples. Only those with significant expression in at least one liver sample (C_T_ < 30) were further analyzed. Four databases were searched for miRNA potentially involved in post-transcriptional regulation of *SLC16A1* expression: MIRTarBASE (mirtarbase.mbc.nctu.edu.tw access on 15th January 2020), microRNA (www.microrna.org access on 15th January 2020), miRDB (www.mirdb.org access on 15 January 2020, score > 60), and Targetscan (targetscan.org access on 15 January 2020, all described as “conserved”).

### 4.5. Measurement of L-Lactate Content in Tissue Samples

Lactate concentration was measured using the colorimetric L-Lactate Assay Kit (Abcam, Cambridge, United Kingdom). In brief, frozen tissue samples (10 mg) were resuspended in five volumes of the supplied Lactate Assay Buffer using Dounce homogenizer, subsequently deproteinized with Deproteinizing Sample Preparation Kit (Abcam), and processed according to protocol supplied by kit manufacturer. Absorbance was measured at 450 nm with Infinite Pro 200 plate reader (Tecan, Männedorf, Switzerland). Lactate concentration was determined based on a standard curve, freshly prepared prior to absorbance measurement.

### 4.6. pH Measurement

pH measurements in tissue samples were performed with s FiveGo F2 pH meter equipped with an InLab Ultra-Micro-ISM electrode for measurements in micro volumes (Mettler Toledo, Greifensee, Switzerland). Frozen tissue samples (20 mg) were homogenized in liquid nitrogen and resuspended in five volumes of ultra-pure deionized water, and pH was measured after 20 min of incubation in room temperature.

### 4.7. Immunohistochemistry

Immunohistochemistry was used to confirm the presence of MCT1, previously determined by mass spectrometry (MS), and to estimate its cellular localization. Three representative liver samples were selected from each liver pathology studied, as well as control livers. Tissue samples were fixed in buffered 10% formalin and embedded in paraffin. Mouse monoclonal MCT1 antibody was used (sc-365501, Santa Cruz Biotechnology, United States; dilution 1:100; incubation time 30 min). The slides were immunostained using a Dako EnVision FLEX+ visualization system with an automated immunostainer (Dako Autostainer Link 48) according to the manufacturer’s instructions. The reaction was developed with a diaminobenzidine substrate–chromogen solution and counterstained with hematoxylin.

### 4.8. Statistical Analysis

Normality of distribution of quantitative variables was determined by means of a Shapiro–Wilk test. Statistical difference (*p* < 0.05) between groups was determined by the non-parametric Kruskal–Wallis test for multiple comparisons with a post-hoc Dunn’s test, and correlations with Spearman rank test. The calculations were based on all samples, substituting undetectable protein concentrations (lower or equal to 0.1 nmol/L,) with zero. For multiple miRNA/protein correlation analyzes, Bonferroni correction was finally applied by multiplying the obtained *p*-values by the number of analyzed miRNAs. The statistical calculations were performed using Statistica 13.3 Software Package (TIBCO Software Inc, Palo Alto, CA, USA).

## Figures and Tables

**Figure 1 ijms-21-01606-f001:**
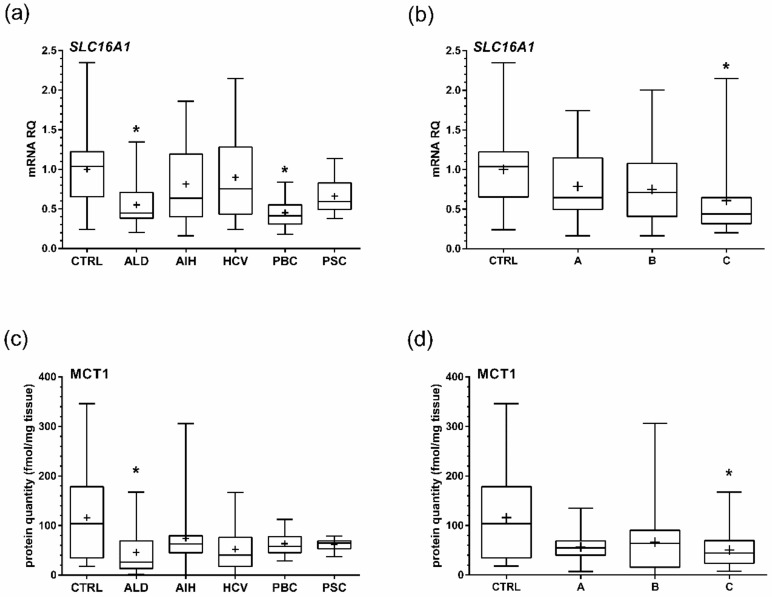
Relative quantity of (**a**,**b**) *SLC16A1* mRNA and (**c**,**d**) monocarboxylate transporter 1 (MCT1) protein in the studied liver tissue samples. The mean expression of five house-keeping genes was used as a reference for mRNA quantification (*PPIA*, *RPLP0*, *RPS9*, *ACTB*, and *HMBS*). Data is presented for different liver pathologies: hepatitis C (HCV), primary biliary cholangitis (PBC), primary sclerosing cholangitis (PSC), alcoholic liver disease (ALD), autoimmune hepatitis (AIH), and control livers—normal tissue from metastatic colon cancer livers (CTRL) (**a**,**c**)—as well as for different Child–Pugh stages of liver disease (A, B, C) (**b**,**d**). * *p* < 0.05 (Kruskal-Wallis test for multiple comparisons with post-hoc Dunn’s test) in comparison to the controls.

**Figure 2 ijms-21-01606-f002:**
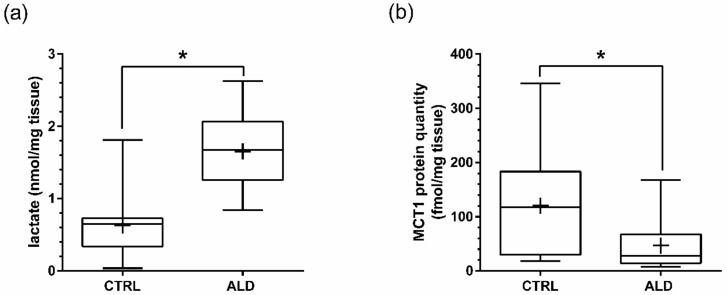
Lactate concentration (**a**) and MCT1 protein abundance (**b**) in the control (CTRL) and alcoholic liver disease (ALD) samples. All measurements were performed in homogenized tissue samples. * *p* < 0.05 (Mann-Whitney U test) in comparison to the controls.

**Figure 3 ijms-21-01606-f003:**
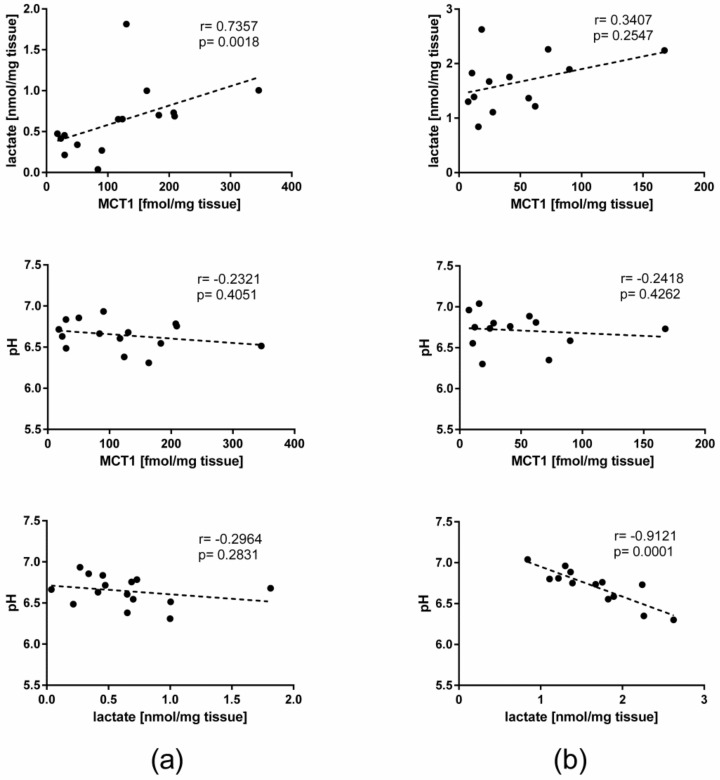
Analysis of correlation between lactate tissue concentration, pH, and MCT1 protein abundance in the control (**a**) and alcoholic liver disease (**b**) samples. All measurements were performed in homogenized tissue samples.

**Figure 4 ijms-21-01606-f004:**
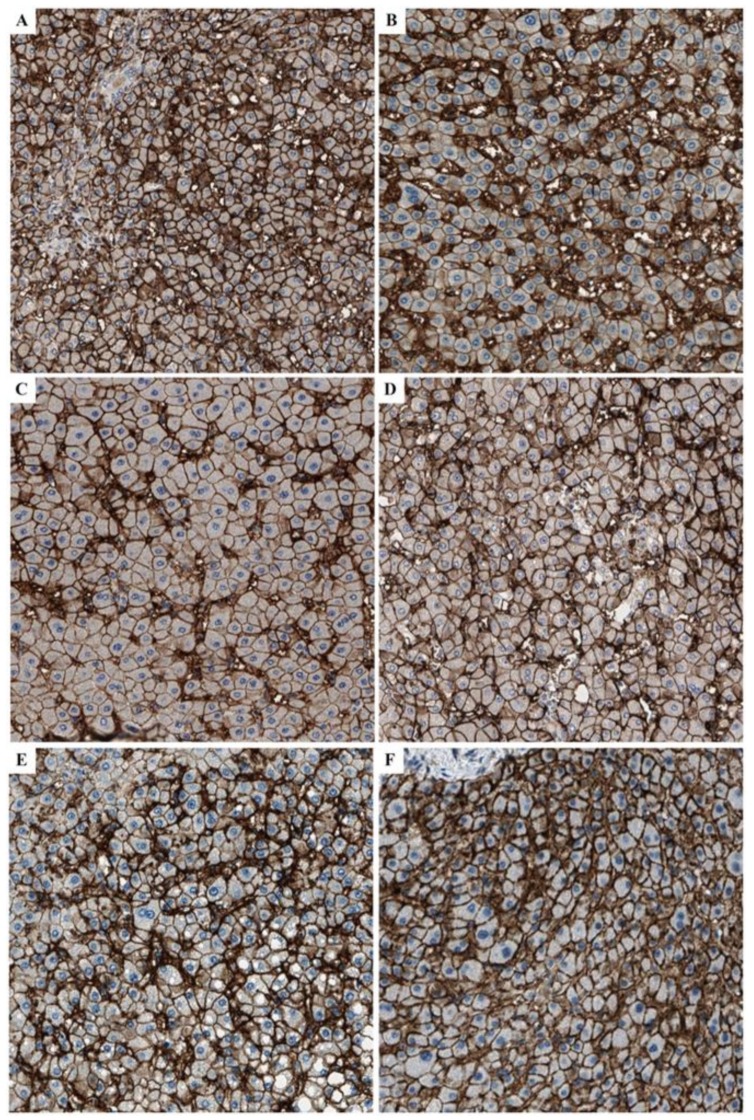
Positive membranous expression of MCT1 in (**A**) autoimmune hepatitis, (**B**) alcoholic liver disease, (**C**) primary biliary cirrhosis, (**D**) primary sclerosing cholangitis, (**E**) hepatitis C-related liver cirrhosis, and (**F**) normal liver.

**Table 1 ijms-21-01606-t001:** Correlation (Spearman coefficient *r*) between selected miRNA and MCT1 tissue content.

miRNA	CTRL (*n* = 20)	Liver Pathologies (*n* = 77)	HCV (*n* = 21)	PBC (*n* = 10)	PSC (*n* = 6)	ALD (*n* = 20)	AIH (*n* = 20)	All Samples (*n* = 97)
hsa-miR-27a-3p	**−0.591**	**−0.251**	−0.156	−0.417	**−0.829**	0.174	−0.335	**−0.301 ***
hsa-miR-27b-3p	−0.107	0.003	0.339	0.150	−0.200	−0.039	0.102	0.092
hsa-miR-29a-3p	−0.128	−0.095	0.042	−0.533	−0.543	0.223	−0.137	0.020
hsa-miR-29b-3p	−0.152	−0.031	−0.027	−0.100	−0.486	0.167	0.135	−0.163
hsa-miR-29c-3p	−0.099	−0.184	−0.129	**−0.733**	−0.543	0.062	**−0.480**	−0.186
hsa-miR-145-5p	**−0.475**	−0.166	−0.005	0.533	−0.543	0.069	−0.346	**−0.253**
hsa-miR-320a-3p	−0.209	−0.053	−0.048	0.333	−0.714	0.463	−0.368	0.049
hsa-miR-324-5p	−0.412	−0.123	−0.223	0.583	−0.486	0.089	−0.348	−0.160
hsa-miR-342-3p	0.015	**−0.269**	−0.252	0.250	−0.314	−0.323	−0.280	**−0.310 ***
hsa-miR-374a-5p	−0.379	−0.234	−0.323	−0.617	−0.486	0.107	−0.437	**−0.312 ***
hsa-miR-374b-5p	−0.436	−0.195	−0.075	−0.100	−0.600	0.045	**−0.628**	−0.173
hsa-miR-376a-3p	−0.131	−0.182	−0.319	0.317	−0.486	−0.003	−0.391	−0.145
hsa-miR-425-5p	−0.286	−0.139	0.000	0.200	−0.600	−0.018	−0.234	−0.045
hsa-miR-495-3p	0.083	−0.219	−0.229	0.350	−0.257	−0.077	**−0.503**	−0.193
hsa-miR-590-3p	−0.189	−0.126	−0.209	**−0.833**	−0.086	0.045	−0.292	−0.124
hsa-miR-605-5p	−0.042	−0.198	0.065	−0.167	−0.371	**−0.531**	−0.257	−0.090

Significant negative correlations (*p* < 0.05) are marked with bold font; * correlation significant after application of Bonferroni correction.

**Table 2 ijms-21-01606-t002:** Characteristics of the study subjects.

Parameter/Disease	Controls *n* = 20	HCV *n* = 21	PBC *n* = 10	PSC *n* = 6	ALD *n* = 20	AIH *n* = 20
Sex (male/female)	11/9	10/11	1/9	4/2	16/4	8/12
Age (years)	63 ± 10	52 ± 5	59 ± 4	43 ± 10	51 ± 6	47 ± 16
Child–Pugh (A/B/C)	-	7/10/4	2/4/4	3/3/0	0/8/12	6/6/8
Total bilirubin (mg/dl)	0.59 ± 0.25	2.38 ± 1.37	6.42 ± 6.72	8.14 ± 8.14	4.4 ± 4.02	3.54 ± 3.53
Albumin (g/dl)	3.89 ± 0.38	3.31 ± 0.45	3.13 ± 0.65	3.7 ± 0.44	3.03 ± 0.50	3.29 ± 0.39
PT (s)	12.7 ± 2.3	14.4 ± 2.0	12.5 ± 1.2	13.2 ± 2.8	16.0 ± 2.2	14.6 ± 2.5
INR	1.14 ± 0.21	1.39 ± 0.27	1.19 ± 0.21	1.4 ± 0.52	1.47 ± 0.23	1.42 ± 0.41

HCV: hepatitis C, PBC: primary biliary cholangitis, PSC: primary sclerosing cholangitis, ALD: alcoholic liver disease, AIH: autoimmune hepatitis, PT: prothrombin time, and INR: international normalized ratio. Mean values and standard deviation values are given for quantitative variables.

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
