# Peer review of "Monocarboxylate Transporter 1 (MCT1) in Liver Pathology"

_ijms, 2020, doi:10.3390/ijms21051606_

Round 1
Reviewer 1 Report
In this manuscript Drozdzil et al. determines the differential expression of the MCT1 transporters between liver hepatic diseases and healthy livers. Authors described accurately the importance of this transporters family in several diseases and they found a downregulation of MCT1 in ALD.
Comments:
Figure 1
Authors should indicate if 'control samples' are healthy livers or if are the surrounding tissues of unhealthy livers.
1a, authors should indicate in the figure the housekeeping gene used to analyse the RT-qPCR results.
1a,b authors should divide figure 1a and 1b in two different graphs.
microRNAs
It is not entirely clear how microRNAs suggested by authors could regulate SLC16A1 transcript. Are these microRNAs targeting a target site on SLC16A1 mRNA or is it just an epiphenomenon?
Figure 2
Authors should indicate if they are analysing intracellular or extracellular Lactate concentration. Again, authors should divide graph in two distinct images as it makes data confusing.
Figure 3
Authors should indicate if they are analysing intracellular or extracellular pH.
Figure 4
Authors claimed that expression of MCT1 was found in all studied liver pathologies and control samples by IHC. Authors should then quantitative determine MCT1 protein expression from IHC that it is more accurate compared to MS analysis. Authors should then analyse again all data using IHC protein analysis.
Author Response
Responses to the Reviewer 1
Figure 1 Authors should indicate if 'control samples' are healthy livers or if are the surrounding tissues of unhealthy livers.
Response: This information has been added to the figure’s description (“control livers – normal tissue from metastatic colon cancer livers”)
1a, authors should indicate in the figure the housekeeping gene used to analyse the RT-qPCR results.
Response: This information has been added to the figure’s description (“Mean expression of five house-keeping genes was used as reference for mRNA quantification (PPIA, RPLP0, RPS9, ACTB and HMBS)”
1a,b authors should divide figure 1a and 1b in two different graphs.
Response: Figure has been divided, now there are two separate graphs for mRNA analysis (a, b) and two separate graphs for protein quantification (c, d).
microRNAs
It is not entirely clear how microRNAs suggested by authors could regulate SLC16A1 transcript. Are these microRNAs targeting a target site on SLC16A1 mRNA or is it just an epiphenomenon?
Response: microRNAs analyzed in the current paper were predicted to target SLC16A based on in silico modeling, and it is stated in Methods: “Four databases were searched for miRNA potentially involved in post-transcriptional regulation of SLC16A1 expression: MIRTarBASE (mirtarbase.mbc.nctu.edu.tw), microRNA (www.microrna.org), miRDB (www.mirdb.org, score > 60), and Targetscan (targetscan.org).” As that must be considered as one of the study limitations, it is stated in Discussion that “All the obtained miRNA data must be treated as preliminary, as no functional validation of miRNA involvement in regulation of SLC16A1 expression was performed in the present study and further evidence must be obtained from additional in vitro experiments.”
Figure 2 Authors should indicate if they are analysing intracellular or extracellular Lactate concentration. Again, authors should divide graph in two distinct images as it makes data confusing.
Response: Figure 2 has been divided in two separate graphs (a, b). Lactate concentration was measured in homogenized tissue samples, and it is now stated in figure’s footage.
Figure 3 Authors should indicate if they are analysing intracellular or extracellular pH.
Response: Similarly to lactate concentration, pH measurements were performed in homogenized tissue samples, so they are not intra- nor extracellular measurements. Adequate notice has been added to figure’s footage.
Figure 4 Authors claimed that expression of MCT1 was found in all studied liver pathologies and control samples by IHC. Authors should then quantitative determine MCT1 protein expression from IHC that it is more accurate compared to MS analysis. Authors should then analyse again all data using IHC protein analysis.
Response: Actually, IHC analysis was not performed in all the studied samples. Three available liver samples were selected from each liver pathology and control livers, and subsequently used for immunostaining. IHC was used to confirm the presence of MCT1 determined by MS, and to estimate its cellular localization. The adequate statement has been added to Methods section. Since IHC was done only in single (available paraffin embedded samples) samples, quantitative analysis of staining intensity in all LC/MS-MS quantified samples was not possible.
Reviewer 2 Report
This study focuses on the expression of SLC16A1 in liver pathologies. Overall I find the paper to be scientifically sound. Although the significance of SLC16A1 in liver pathologies is unclear, I believe this study still merits publication. The manuscript is well written and clear, I would only recommend minor editing for spelling and grammar. The presentation of figures is clear and understandable.
Author Response
Responses to the Reviewer 2
The manuscript is well written and clear, I would only recommend minor editing for spelling and grammar.
The manuscript was subjected to editing for spelling and grammar (corrections are marked in the revised version of the manuscript).
Round 2
Reviewer 1 Report
Authors amended the manuscript as requested and replied to all other comments. Paper is accepted in the current form.